# Kin recognition: Neurogenomic response to mate choice and sib mating avoidance in a parasitic wasp

**Aurore Gallot** [1] *, **Sandrine Sauzet**[1,2], **Emmanuel Desouhant**[1]

**1** Laboratoire de Biométrie et Biologie Evolutive, CNRS, Université Lyon 1, Université de Lyon, UMR 5558, Villeurbanne, France, **2** Institut de Génétique Humaine, CNRS–Université de Montpellier, UMR 9002, Biology of Repetitive Sequences, Montpellier, France

* aurore.gallot@univ-lyon1.fr

**Data Availability Statement:** Raw sequencing data, head transcriptome data, the expression raw count table per gene (HTSEQ output) and associated metadata are available in the NCBI Gene Expression Omnibus (GEO) repository and

## Abstract

Sib mating increases homozygosity, which therefore increases the risk of inbreeding depression. Selective pressures have favoured the evolution of kin recognition and avoidance of sib mating in numerous species, including the parasitoid wasp *Venturia canescens*. We studied the female neurogenomic response associated with sib mating avoidance after females were exposed to courtship displays by i) unrelated males or ii) related males or iii) no courtship (controls). First, by comparing the transcriptional responses of females exposed to courtship displays to those exposed to controls, we saw a rapid and extensive transcriptional shift consistent with social environment. Second, by comparing the transcriptional responses of females exposed to courtship by related to those exposed to unrelated males, we characterized distinct and repeatable transcriptomic patterns that correlated with the relatedness of the courting male. Network analysis revealed 3 modules of specific 'sib-responsive' genes that were distinct from other 'courtship-responsive' modules. Therefore, specific neurogenomic states with characteristic brain transcriptomes associated with different behavioural responses affect sib mating avoidance behaviour.

## Introduction

Mate choice often depends on individual attractiveness. In many species, females discriminate between competing males [1]. Female mate choice sometimes occurs after elaborate courtship displays, and males are selected according to their highly heritable qualities ('good genes' hypothesis) and/or their genetic compatibility with the female ('compatible genes' hypothesis) [2]. The 'good genes' hypothesis predicts that females favour reproduction with males carrying traits that are honest indicators of good genes or as a result of sensory bias [3, 4], hence obtaining genetic benefits for their offspring [5]. The 'compatible genes' hypothesis predicts that offspring fitness is correlated with genetic dissimilarity between mating partners, where each female prefers males that possess genes compatible with their own genotypes. Sib mating avoidance fits into this second hypothesis [6, 7]; in this case, kin recognition is crucial for enabling females to discriminate between related and unrelated mates. Sib mating avoidance

accessible through accession number GSE124129. Normalized expression dataset, matrix reporting the experimental design, and the script used to analyzed dataset are available as Supporting Information. A full list of differentially expressed genes and associated statistics is available in S2 and S3 Tables. Additional data may be requested from the authors.

**Funding:** This work was funded by the Agence Nationale de la Recherche (https://anr.fr/en/) JCJC AVOIDINBRED (ANR-17-CE02-0004-01) attributed to AG. The funders had no role in study design, data collection and analysis, decision to publish, or preparation of the manuscript.

**Competing interests:** The authors have declared that no competing interests exist.

has been extensively documented in numerous animal species (including species that are social and solitary) and reduces the risk of inbreeding depression [8].

In colonial marine invertebrates, contact between 2 conspecific colonies may trigger fusion if the 2 colonies are identical or related, or aggressive rejection if the 2 colonies are not related. Molecular bases for kin recognition involve several highly polymorphic loci that have been well characterized in 2 species: the cnidaria *Hydractinia symbiolongicarpus* (reviewed in [9]) and the chordata *Botryllus schosseri* (reviewed in [10]). In terrestrial animals, most of the presently identified mechanisms usually rely on individual-specific olfactory cues and sensory organs able to discriminate these cues from one another. In vertebrates, major histocompatibility complex genes confer individual olfactory identities [11–14]. These olfactory cues bind to receptors often located on the vomeronasal organ neurons and are used by females to evaluate mate relatedness [15–19]. In insects, numerous species discriminate kin by using odour differences inherent to divergence in the cuticular hydrocarbon composition [20–23]. Odour molecules bind to specific sensory receptors located on sensilla basiconica, where dozens of olfactory sensory neurons project to the antennal lobes [24–26], where olfactory second-order neurons project to the lateral horn and mushroom bodies [27, 28]. To the best of our knowledge, no genome-wide study has addressed how kin recognition signals are processed in the context of mate choice and sib mating avoidance. Transcriptomic studies provided recent insight into female mating decisions [29]. Coordinated changes in the expression of many genes in female brains, *i.e.*, a neurogenomic response, have been identified following courtship in Poeciliidae fishes [30–32]. This response depends on male attractiveness and is in accordance with female preferences.

Hymenopterans (including bees, ants, wasps and sawflies) are relevant models to explore the biology of mate choice, especially regarding sib mating avoidance and its underlying biochemical mechanisms. These insects share an ancestral and unusual sex determination system called single-locus complementary sex determination (sl-CSD), where sex is determined by heterozygosity at the sl-SCD locus. Heterozygous individuals develop as females, hemizygous individuals develop as males, but homozygous individuals develop into diploid unviable or sterile males [33–36]. When sib mating occurs, the risk of genetic incompatibility between partners sharing the same allele at the sl-CSD locus is one in two; in that case, half of the progeny will be homozygous diploid sterile males (with notable exceptions [37, 38]). Therefore, the sl-CSD has an additional immediate effect on inbreeding. Consequently, Hymenoptera are much more exposed to inbreeding depression than any other diploid species [39]. In this context, congruent with the compatible gene hypothesis, one expects that selection will favour females that prefer males with a distinct allele at the sl-CSD locus, thus excluding sibs. In the parasitoid wasp *Venturia canescens*, which has sl-CSD [40], females only mate once [41], making mate choice particularly decisive. Indeed, in this species, females are able to discriminate kin and non-kin during male courtship based on olfactory-mediated cues [42]. In laboratory conditions, the proportion of successful mate when a single female was in presence of 2 unrelated males was higher (79% of success) than in presence of related males (*i.e.* with 2 brothers) (41% of success) [42]. Furthermore, the latency to mate increased when the courting male was related [42]. Experiments involving choice between 2 males, one brother and one unrelated present in a same area have shown that female mates indifferently with the 2 males [42]. Hence, the two distinct social contexts (presence of a conspecific with and without relation) provoke two contrasting behavioural responses that correspond to the definition of behavioural states, *i.e.*, the performance of a distinct and quantifiable behaviour for a measurable period of time [43].

We tested whether these contrasting behavioural responses to relatedness were correlated to transcriptomic changes in *V. canescens* females. We designed a behavioural experiment

where females were exposed for 10 minutes to an unrelated male or a related male (a brother) or were without any social contact. Then, we used RNA sequencing to compare gene expression profiles from the entire heads of these females. This timing coincided with a period of active courtship, where female wasps perceived and evaluated males, experienced change in receptivity and decided whether or not to mate [44]. Previous studies demonstrated that this temporal frame is adequate to detect early transcriptional changes following mate exposure in different species [30, 45, 46]. We focused on the whole head, where sensory signals are processed and integrated to mediate complex decision making, including mating decisions.

We hypothesized that information processing during male courtship display and mate choice were related to gene expression changes. First, we examined if courtship perception was mediated by changes in gene expression. In this case, we expected the transcriptomic profiles of all courted females would be analogous, whatever the relatedness of the courting male. Then, we assessed whether the differences in female receptivity were associated with gene expression changes. Under this hypothesis, the expectation was that non-receptive females, *i. e.*, solitary females and females courted by related males, would have similar transcriptomic profiles.

## Materials and methods

### Biological model

*Venturia canescens* G. (Hymenoptera: Ichneumonidae) is a solitary parasitic wasp; a maximum of one adult emerges from one host, regardless of the number of parasitoid eggs initially laid in the host. In the wild, *V. canescens* females parasitize pyralid moth larvae feeding on dried fruits, such as figs, carobs, almonds, dates or loquats [47]. Our knowledge of how mating partners encounter each other in the field is largely incomplete as a consequence of the small size of the species, which renders observations difficult. Virgin *V. canescens* females emit chemicals that, in combination with host kairomones, attract males [42]. In turn, males do not attract virgin females at a distance [41]. *V. canescens* females are monandrous and only mate once in their lifetime [41]. Conversely, males can mate more than once. These observations led us to consider that females choose their mate, and we thus focused our study on female transcriptomic responses to courtship. Moreover, females recognize sibs on the basis of a chemical signature emitted by males and can avoid mating with their brothers through kin recognition [42]. This kin recognition ability has also been robustly established in this species in the context of host choice, since females prefer to lay their eggs in hosts already parasitized by conspecifics rather than in hosts already parasitized by themselves [48]. In addition, males also have the ability to distinguish between non-sib and sib females by using chemical markers emitted by females [49].

### Behavioural manipulations

Wild parasitic wasps were collected *via* a large field sampling during the summer of 2014 in orchards in southern France (Valence, N 44˚58'21" E 4˚55'39"). Wasps were maintained on the host *Ephestia kuehniella* (Lepidoptera: Pyralidae) in a climatic chamber (25˚C, 60% humidity, DL 12:12) and fed with a 50/50 honey/water solution. Families were produced as described by Metzger et al. (2010). A total of 90 newly emerged virgin females were individually placed in a box and randomly submitted to one of the three following conditions for a maximum duration of 10 minutes: 1) a compatible male (*i.e.*, unrelated male), 2) an incompatible male (*i. e.*, a brother), or 3) isolated (controls). The experimental design has been conceived in order to minimize the influence of circadian rhythm and genetic background on the results. Newly emerged females were isolated every morning and numbered. A random draw was then made

to establish the order of passage of the females and the treatment assigned. Behavioural experiments took place in the afternoon between 1:00 and 4:00 p.m. for all females that were captured in the morning in the order established by the random draw. A maximum of 2 females from each family have been kept daily, and randomized in one condition. For each given condition, the ten females belonged to different families to avoid an effect of genetic homogeneity on transcriptomic results. New families have been produced for every biological replicate. The 10 minutes period coincides with an active male courtship behaviour [44]. For each female under conditions 1 and 2, one sequence of active courtship parade was observed within the 10 minutes following the introduction of the male to the box for all pairs. Whenever this behaviour sequence was not observed, the pair was eliminated from the study. After this time, or within 3 seconds of mounting, the female and male were separated. For each condition, females were caught in a small tube (3 x 7 cm) and were immediately anaesthetized with $CO_2$ and decapitated with a scalpel. Antennae were removed from the head to focus on the cerebral transcriptome of the decision-making centre rather than the sensory centre. The head is a heterogeneous tissue divided into many functionally distinct regions that could affect inferences about the functional significance of gene expression patterns [50]. However, the study of more specific brain regions remains challenging given the small size of the wasps. Collected heads were immediately flash-frozen and stored at -80°C until RNA extraction. A total of 10 individual heads were collected per experimental condition and pooled to collect higher RNA amounts and to average the expression state across individuals, thus mitigating individual variability during transcriptome comparison [51]. The experiment was sequentially repeated 3 times to obtain biological replicates.

## RNA extraction, library preparation and sequencing

Total RNA extraction was performed in one batch on the 9 biological samples collected (3 experimental conditions x 3 biological replicates). Mechanical cell lysis was performed with metallic beads added to frozen microtubes and shaken with a TissueLyser (Qiagen, 25 hertz 45 seconds). RNA was extracted using Qiazol and an RNeasy mini kit according to the manufacturer's protocol (Qiagen), with optional DNase treatment (Thermo Fisher Scientific). The quality and quantity of total RNAs were assessed on a Bioanalyzer (Agilent) and a Qubit fluorometer (Life Technologies). Polyadenylated RNAs were enriched from 1 μg of high-quality total RNAs with oligo-dT magnetic beads, fragmented and converted to cDNAs (Illumina TruSeq Stranded mRNA Library Prep kit). After adding an A to the 3' end of each cDNA, adapters were ligated, and fragments were amplified by PCR to generate DNA colonies. Each library was labelled, multiplexed and pooled for sequencing on a HiSeq 2500 Illumina sequencer (Fasteris, Switzerland), with a paired-end protocol (2x125 bp). For each of the 9 biological samples, a second library was prepared independently, sequenced and considered a technical replicate. Overall, a total of 18 libraries were sequenced (3 experimental conditions x 3 biological replicates x 2 technical replicates). A minimum of 18 million paired-reads were obtained per library, representing a final dataset of more than 535 million paired-reads.

## RNA-seq quality control, mapping, transcriptome assembly and annotation

The *V. canescens* genome has been sequenced and is currently available (http://bipaa.genouest. org/sp/venturia_canescens/, v.1.0) but thus far, RNA-seq has not been performed on heads [52, 53]. To include all loci detected in the head transcriptome that might be missing from the current annotation, we constructed a *de novo* transcriptome assembly using the genome reference with the TopHat-Cufflinks pipeline [54] (v2.2.1). The global quality of sequences was

assessed with FastQC-0.10.1. Low-quality bases at the ends of reads were trimmed using Trimmomatic-0.36. Adaptor-containing reads, low quality reads (scores phred <30) and N-containing reads were filtered out. Curated reads were then processed through the TopHat-Cufflinks pipeline. First, paired-reads of each sample were aligned to the *V. canescens* genome. TopHat used the Bowtie algorithm to align reads to the genome. Unmapped reads were cut into segments that aligned far apart from one another (between 100 bp to several hundred kb) to predict potential intron/exon structures. Next, an index of hypothetical splice sites was built without any prior information. Each site had to be confirmed by several read segments consistently showing the same alignment pattern. TopHat was used with each of the 18 libraries as input and with the following parameters: stranded libraries (first strand), 5 mismatches/indels allowed, and report the best alignment possible for each read. After mapping, resulting alignment files were provided to Cufflinks, which generated a transcriptome assembly for each sample annotated into genes, transcripts and isoforms. Then, the 18 transcriptomes were merged into one master transcriptome with Cuffmerge. This final transcriptome assembly contained a total of 16,752 genes and provided a uniform basis for calculating gene expression in each condition. The transcriptome was then annotated to gain insight into the functions of the transcripts and the proteins they encode. Sequence similarity was searched for each of the 16,752 predicted genes by comparing the six-frame translation putative products of the nucleotide sequences using BLASTX (v2.2.29+) against the NCBI non-redundant protein database. Transcriptome analysis was completed using gene ontology (GO) annotation, which associated genes with functions in 3 categories: molecular function (molecular activities of gene products), cellular component (where gene products are active) and biological process (pathways and larger processes made up of the activities of multiple gene products). This classification enabled functional interpretation of a large group of genes via enrichment analysis.

## Sample-based clustering and PCA

To obtain an initial overview of gene expression patterns across samples, multivariate analyses were performed on the 18 transcriptomic libraries, including 3 biological replicates for each of the 3 conditions representing a total of 9 samples, each one represented by 2 technical replicates. Those 18 transcriptomes were represented by isolated females (6), females courted by unrelated males (6) or females courted by brothers (6). A raw count table was obtained by using HTSeq [55] (v0.5.4p1). The gene model produced with Cuffmerge was combined with the 18 mapping files previously obtained with TopHat for each gene in each library. This final dataset was exported to R (v3.4.3) [56] for downstream statistical analysis with DESeq2 [57] (v1.30.0). Counts were normalized with the variance stabilizing transformation method (VST), which produced transformed data on the log2 scale and normalized data with respect to library size. The overall variation of expression levels among samples was evaluated with a two-dimensional PCA and with hierarchical clustering, both based on the expression of the 500 genes with the highest variance across libraries. A hierarchical clustering dendrogram was produced based on the sample-to-sample Euclidian distance matrix to obtain an overview of the similarities and dissimilarities between samples. Uncertainty in hierarchical clustering was assessed with Pvclust [58] (v2.0–0) using multiscale bootstrap resampling with an approximately unbiased *P*-value to measure statistical support for each cluster (1,000,000 bootstraps; average agglomerative method; correlation method distance). Two technical replicates from one biological sample of females courted by brothers were obvious outliers in the PCA and hierarchical clustering, most likely due to a problem that occurred during sample conditioning and were thus excluded from the analysis (S1 Fig).

## Differential expression, co-expression network and functional enrichment analysis

All pairs of technical replicates were merged before proceeding to differential expression analysis and network analysis as recommended by Love et al. (2014), keeping a total of 8 samples (3 isolated females, 3 females courted by unrelated males and 2 females courted by brothers). To identify genes with different expression patterns across conditions, we performed pairwise comparisons between 1) females courted by related and unrelated males to controls (*i.e.*, isolated females); and 2) females courted by unrelated males to females courted by related males. Differential expression was tested by using negative binomial generalized linear models implemented in the program DESeq2. We tested for differential expression of all transcripts with an average level of expression superior to 10 reads per library ($n = 14,034$). After normalization, within-group variability (*i.e.*, the variability between biological replicates) was modelled for each gene by the dispersion parameter, which describes the variance of counts by sharing information across genes, assuming that genes of similar average expression strength have a similar dispersion. Among the 14,034 genes tested, only 7 presented an outlier status (*i.e.*, an inconsistent expression pattern across the dataset) and were thus excluded from the test, keeping 14,027 transcript in the final reference transcriptome. For the subset of genes that passed the filtering test, a Wald test *P*-value was calculated and finally adjusted for multiple testing [59]. A gene was considered differentially expressed when the false discovery rate (FDR)-adjusted *P*-value was less than 0.01. We did not apply any log fold change threshold.

We performed a weighted gene co-expression network analysis (WGCNA [60], v1.63) to identify subgroups of genes that shared common expression patterns across the experimental conditions and potentially drove the differences in mate choice. WGCNA is a data reduction technique that regroups genes with similar expression patterns into modules of co-expressed genes and tests the correlations between modules and traits. First, log2-transformed and VST-normalized counts were used to construct a gene co-expression network with the blockwise-Modules function. A correlation matrix was computed for the genes, and the correlations were weighted using a power function ß. Then, genes sharing similar patterns of variation across conditions were regrouped using hierarchical clustering and a dynamic tree-cutting algorithm to define modules of co-expressed genes. For our analysis, the parameters used were as follows: maximum block size = 15,000 genes; power (ß) = 10, minimum module size = 30 genes. The remaining parameters were kept at the default settings. A colour name was assigned to each module, and biologically interesting modules were identified by correlating a summary profile for each module to external experimental conditions; *P*-values <0.05 were considered significant and numbered from 1 to 11. Finally, the potential 'hub genes' in every significant module were identified. So-called hub genes may influence the expression of other genes in their module and may be causal factors for a trait of interest. Such hub genes are potentially biologically relevant by driving phenotypic variations [61, 62]. The identification of hub genes relies on both connectivity with other genes from the module and the correlation to the trait. Accordingly, genes were ranked according to their module membership values in each module. The top 5 hub genes of every module were annotated, and their expression pattern, which was representative of the module they belonged to, was detailed.

Functional characterization of gene sets (*i.e.*, genes with differential expression, or modules of co-expressed genes) was analysed using enrichment analysis and gene ontology annotations. An enrichment test was performed on test sets compared to the full transcriptome with Blast2GO [63] (v5.0). The proportion of genes associated with GO terms was compared between the test set and the transcriptome (14,027 transcripts) with a unilateral Fisher's exact test (one-sided), *P*-values < 0.01 were considered significantly enriched.

## Results

### The transcriptomic response to courtship is consistent with the mate preference of females

We first determined whether the male courtship display provoked a transcriptomic response in females. To do so, we built the first *V. canescens* head transcriptome by sequencing more than 500 million reads using a genome guide assembly (18 to 55 million paired-reads from 18 libraries, S1 Table) and used it as a reference to compare the different head transcriptomes of females. After quality filtering a mean of 98.4% of paired-reads were kept, on which 70.8% were successfully mapped to the genome (representing a mean of 20 millions per sample, for a total of 363 million paired-reads, S1 Table). Such values corresponded to the high-quality standards observed in other Hymenopteran species with an annotated genome [64]. The transcriptome constructed with these sequences encompassed 16,752 genes. Overall, 76% of predicted genes get a blast hit (12,740) while 4,012 sequences get no hit. Among genes matching with blast, 89.4% presented their best hit with an insect sequence, of which 84.4% match more specifically to a hymenopteran insect sequences. Finally, most of the genes were uncharacterized, since only 33.4% of the predicted genes (5,589) were associated with at least one Gene Ontology (GO) functional annotation.

Based on this unique reference, head transcriptomes from females 1) courted by an unrelated male, 2) courted by a related male, and 3) isolated (controls) were first analysed without *a priori* knowledge. Principal component analysis (PCA, Fig 1A) revealed that transcriptomes shared higher similarity within a given experimental condition than between different experimental conditions. PCA defined three consistent clusters according to the social environment proposed: the control group where females were kept in isolation, the group of females courted

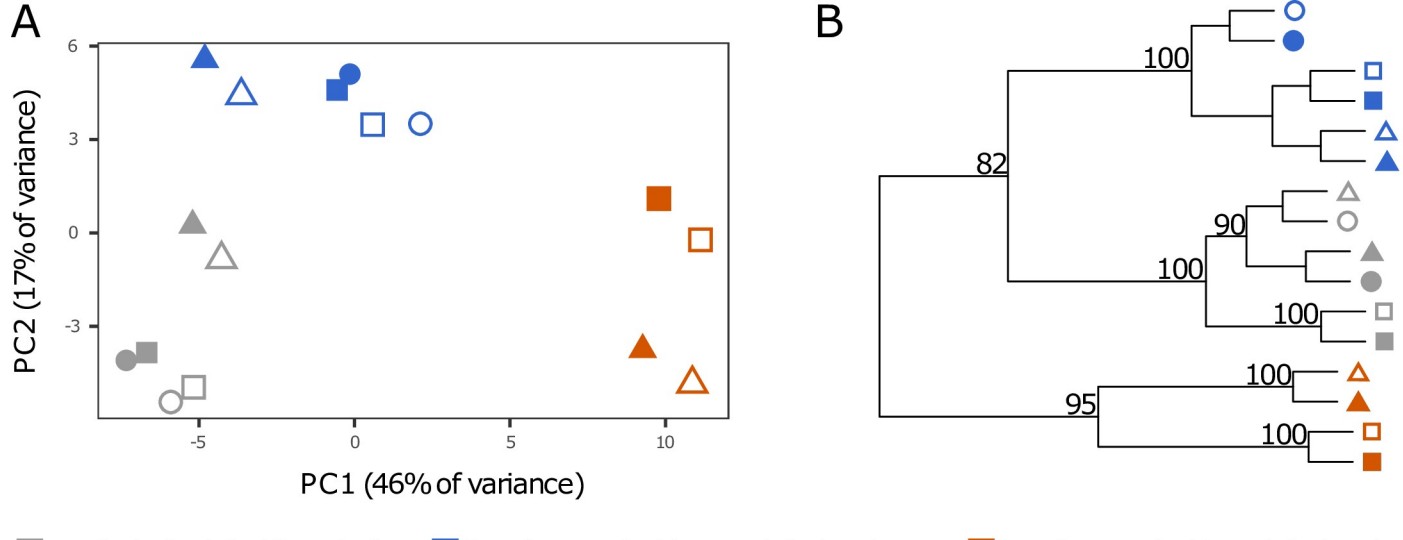

**Fig 1. Multivariate analysis of the 16 RNA-seq libraries based on the gene expression profiles of the 500 genes with the highest variance across samples.** The samples are clustered by the experimental conditions that females were exposed to. Experimental conditions are represented by the three different colours (grey: control; orange: females courted by related males; blue: courted by unrelated males), shapes indicate the three biological replicates, and the filling of the shape shows the 2 technical replicates. (**A**) Two-dimensional principal component analysis (PCA) plot. (**B**) Hierarchical clustering dendrogram based on sample-to-sample distances. Statistical support is indicated by an approximately unbiased *P*-value with one million multiscale bootstrap resamplings (all bootstrap values were > 50%, those < 80% were not shown for clarity).

by unrelated males, and the group of females courted by related males. Principal component 1 separated females courted by related males from the two other conditions and explained the largest fraction of variance in gene expression (46%). Principal component 2 separated females courted by unrelated males from the two other conditions and accounted for 17% of the variance in gene expression (Fig 1A). These results were supported by hierarchical clustering analysis of sample-to-sample distance, which established that samples were clustered by the experimental conditions that they were exposed to (Fig 1B). Bootstrap resampling provided strong statistical support for this result: the control group (isolated females) formed a cluster with 100% support (Fig 1B), while the group of females courted by unrelated males constituted a cluster with 100% support. Finally, females courted by related males constituted a group with 95% support (Fig 1B). Together, these results showed that i) male courtship had an effect on the female transcriptome, and ii) being courted by a related or an unrelated male provoked two distinct transcriptomic responses.

## Functional characterization of differentially expressed genes following courtship

The transcriptomes of females courted by related and unrelated males were compared to those of isolated females to identify the female neurogenomic response to courtship. We identified 1,001 differentially expressed genes (DEGs), representing 7.1% of the 14,027 gene sets tested (Fig 2A, S2 Table). Among the 1,001 DEGs, 463 had higher expression in isolated females (3.3% of total transcriptome), gene ontology enrichment analysis reveals that this set of gene was enriched in DNA-binding Transcription Factor Activity (full list in S2 Table). In contrast, 538 DEGs were overexpressed in courted females (3.8% of total transcriptome), gene ontology enrichment analysis reveals that this set of gene was enriched in Reproductive Behaviour (full list in S2 Table).

Among the DEGs, we noticed three neuropeptides associated with female receptivity, namely, *Capability*, *PDF* and *SIFamide*, were all downregulated in courted females (Fig 2). The orthologue to *muscleblind*, required for normal regulation of female sexual receptivity [65], was also detected among the genes overexpressed in courted females. Notably, we also reported two genes associated with dopamine, *ebony* (downregulated in courted females), *DAT (dopamine transporter*, overexpressed in courted females), and one gene associated with serotonin (*5-HT1A*, *serotonin receptor 1A*, overexpressed in courted females) (Fig 2).

## Relatedness of the courting male influences the female head transcriptome

We compared females courted by unrelated males to those courted by related males to determine whether the relatedness to the courting male had an impact on the female transcriptomic response. By comparing these two groups, we identified 831 DEGs representing 5.9% of the tested genes (Fig 3A, S3 Table). Among these DEGs, 481 genes presented higher expression in females courted by related males (3.4%), gene ontology enrichment analysis reveals that this set of gene was enriched including ATP metabolism and Ribosome (full list in S3 Table). Moreover, 350 DEGs were overexpressed in females courted by unrelated males (2.5%). Enrichment analysis revealed 22 associated GO terms including Reproductive Behaviour and Male Mating Behaviour (full list in S3 Table). These 2 GO terms both refer to the same 2 genes, which belong to the *yellow-major royal jelly protein* family (Fig 3B, S2 Fig), an insect gene family notably associated with behaviour and caste specification [66]. Interestingly, we also noticed the regulation of the transcription factor *PAX6*, which is required for normal brain structure and function, notably locomotory behaviour [67] (Fig 3B).

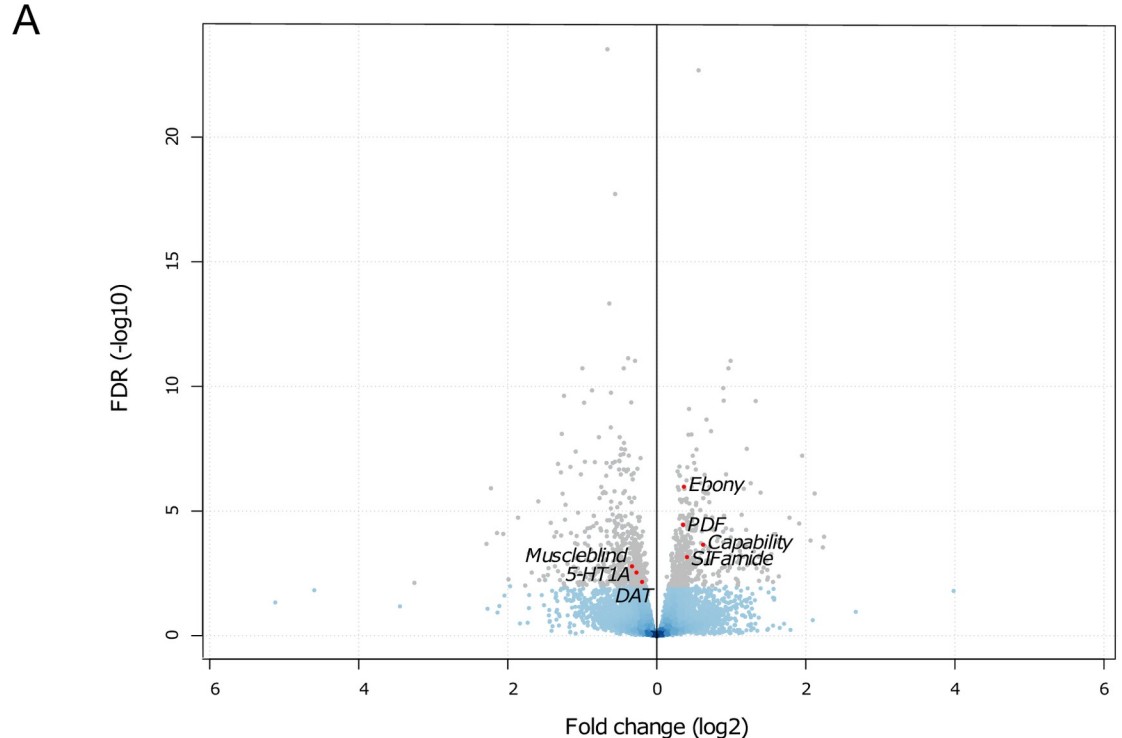

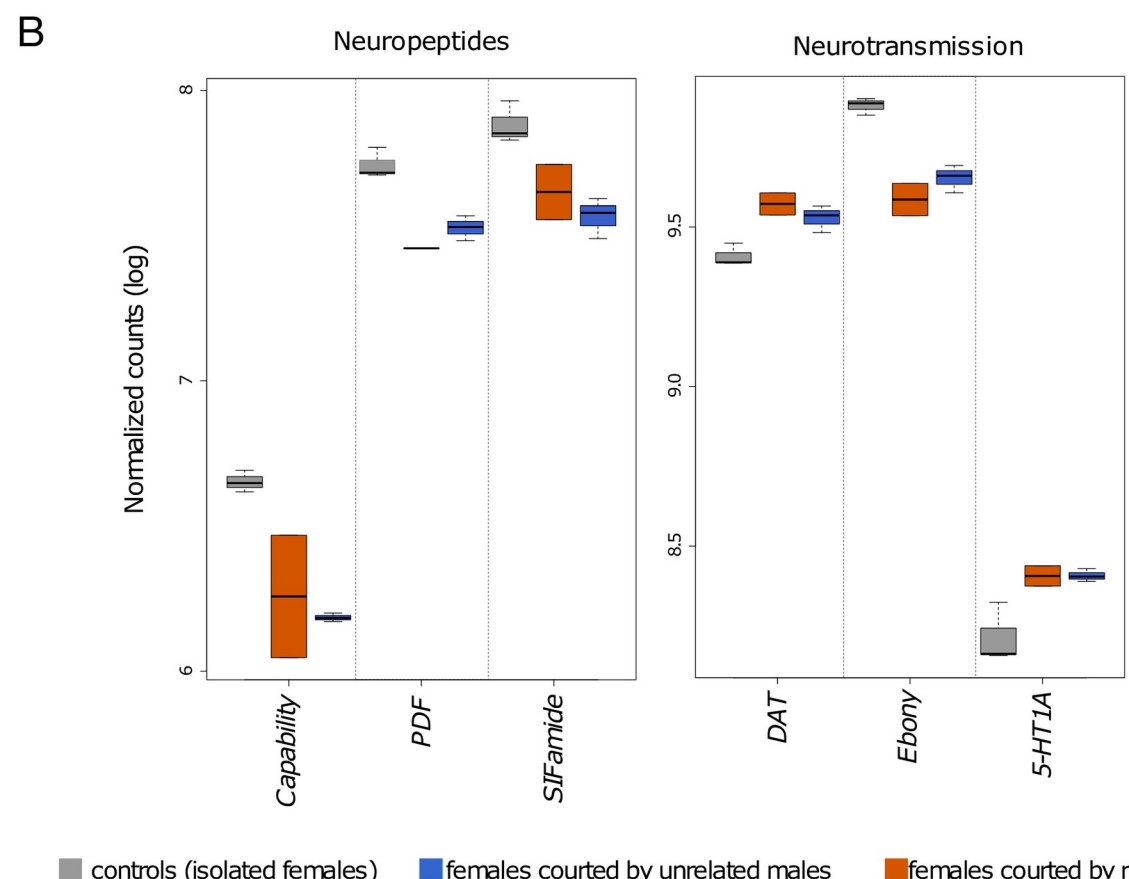

**Fig 2. Transcriptomic response to courtship: 7.1% of the total transcriptome (grey) was regulated in response to courtship (1,001 DEGs). (A)** Comparison of females courted by related and unrelated males to controls revealed 538 genes were overexpressed in courted females (negative fold change values) and 463 genes were overexpressed in isolated females (positive fold change values). (**B**) Boxplot showing significant expression changes following courtship in highlighted genes mentioned in the text. The X axes indicate gene names, the Y axes show normalized counts after log transformation, and the boxplot whiskers show the range of read counts between biological replicates. *PDF*, pigment-dispersing factor; *DAT*, Dopamine transporter; *5-HT1A*, Serotonin transporter 1A.

## Co-expression network and characterization of 'courtship-responsive' modules and 'sib-responsive' modules

To better characterize the variations in gene expression according to the experimental conditions experienced by females, we applied a co-expression network analysis using WGCNA on the 14,027 genes that passed through the expression filter. The gene co-expression network grouped genes that shared a similar expression pattern across different experimental conditions into modules.

Overall, the 14,027 genes were organized into 50 modules of highly correlated genes symbolized by a colour, with sizes varying from 32 to 2,675 genes (Fig 4, Table 1). Among the 50 modules defined by the cluster analysis, 11 modules (numbered from 1 to 11) had significant correlations with at least one of the experimental conditions experienced by the females (Table 1, Fig 4B, S3 Fig); these were considered biologically relevant and were further analysed.

First, three modules were associated with courtship display by an unrelated male (1, 2 and 3), which grouped 1,239 genes. Then, three modules were associated with sib-responsive genes (4, 5 and 6; S3 Fig; S4 Table), which grouped 2,824 genes responding only in the presence of related males. Next, two modules were associated with response to courtship, whatever the degree of relatedness of the courting males (modules 7 and 8; S3 Fig; S4 Table), which grouped 2,780 genes. Finally, three modules were associated with a gene expression pattern peculiar to each of the 3 social environments (9, 10 and 11; S3 Fig; S4 Table).

## Discussion

In the current study, we characterized the female neurogenomic response associated with sib mating avoidance by identifying remarkable differences in the head transcriptome triggered by courtship display that differed according to the relatedness of the courting male. In *V. canescens* females, mate relatedness influences female sexual receptivity and is estimated during male courtship displays through chemical cues [42]. Unrelated males induce sexual receptivity in females, whereas related males induce weak sexual receptivity. Hence, sib mating avoidance can be considered a behavioural state, similar to other transitory behaviours such as aggressiveness [68] or singing [69]. We showed that sib mating avoidance is associated with distinct and repeatable cerebral transcriptomic patterns involving a significant part of the transcriptome (>5%). Such results fit the definition of neurogenomic state, *i.e.*, a distinct and repeatable pattern of gene expression in the brain revealed by contrasting brain transcriptomes of individuals across different behavioural states. Despite the quite low number of biological replicates, the highly contrasted transcriptomes observed in the different social contexts suggest that sib mating avoidance behaviour could be considered a neurogenomic state. This research paves the way for further study on neurogenomic effects of sib mating avoidance in many species where such behaviours have been described and, thus, may contribute to the understanding of the molecular mechanisms underlying the evolution of avoiding consanguinity.

We measured major transcriptomic modifications occurring in the female head following a courtship display. Females exhibited transcriptomic pattern changes following an encounter

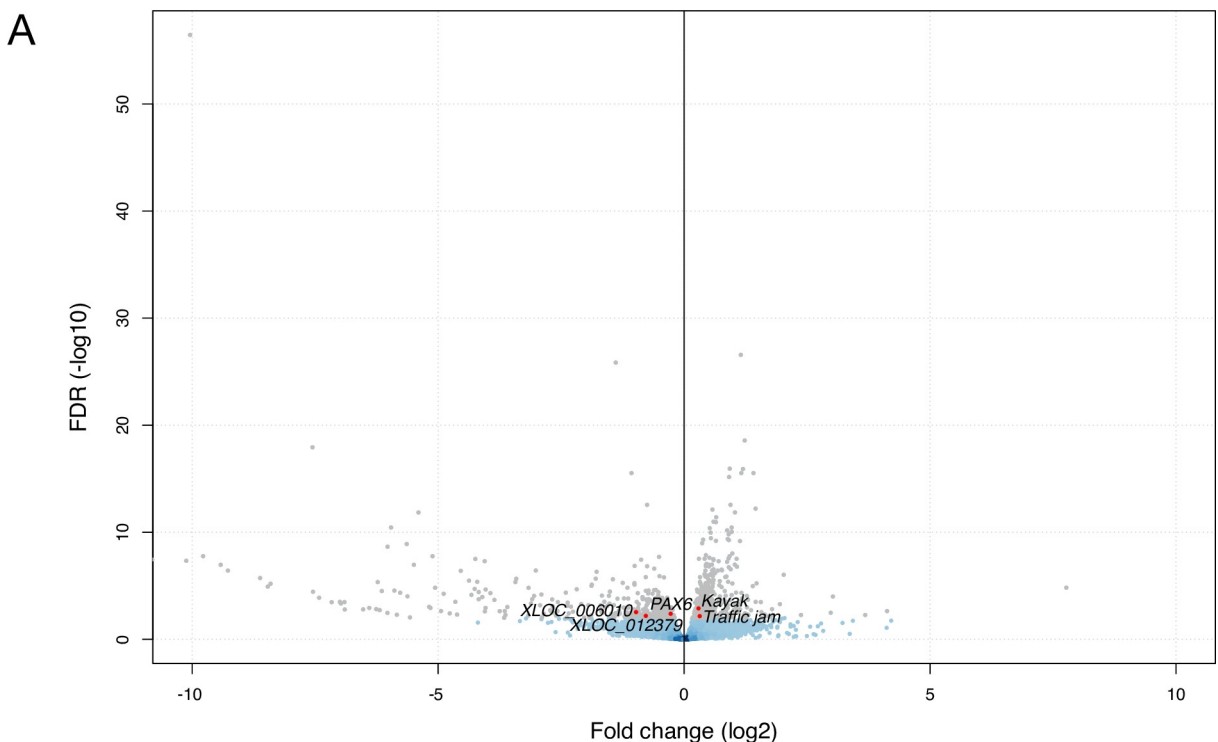

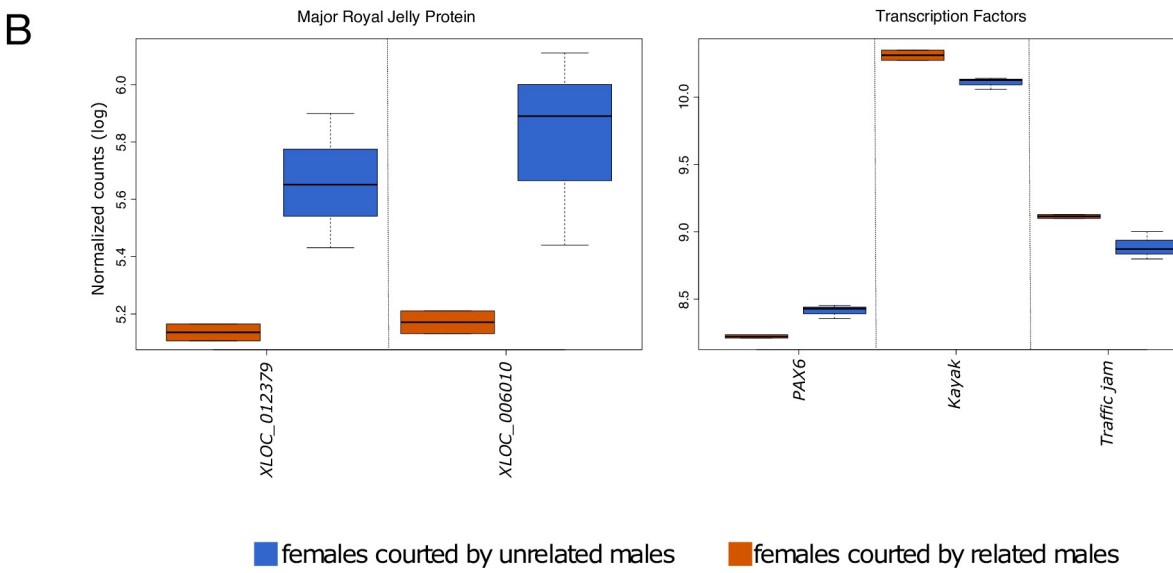

**Fig 3. Influence of relatedness of courting males on females' transcriptome expression: 5.9% of the total transcriptome (grey) was regulated according to the relatedness of the courting male.** (**A**) Comparison of females courted by related males to females courted by unrelated males showed 481 genes overexpressed in females courted by related males (positive fold change values) and 350 genes overexpressed in females courted by unrelated males (negative fold change values). (**B**) Boxplot showing significant expression changes according to the relatedness of the courting male for highlighted genes mentioned in the text. The X axes indicate gene names, Y axes show normalized counts after log transformation, and boxplot whiskers show the range of read counts between biological replicates. *Yellow—mrjp*: yellow major royal jelly proteins; *PAX6*: paired box 6.

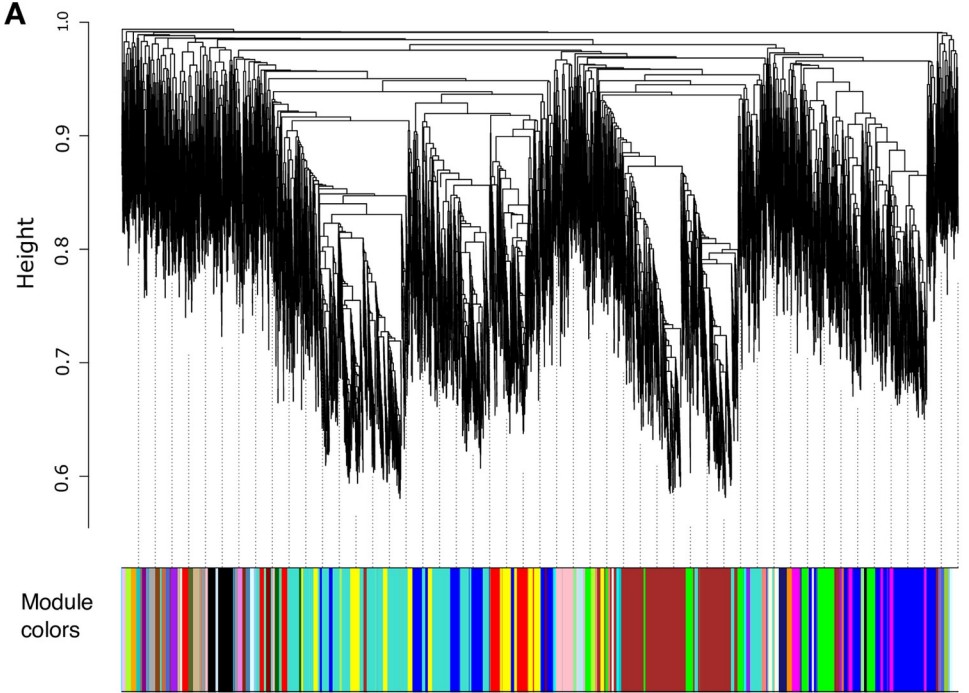

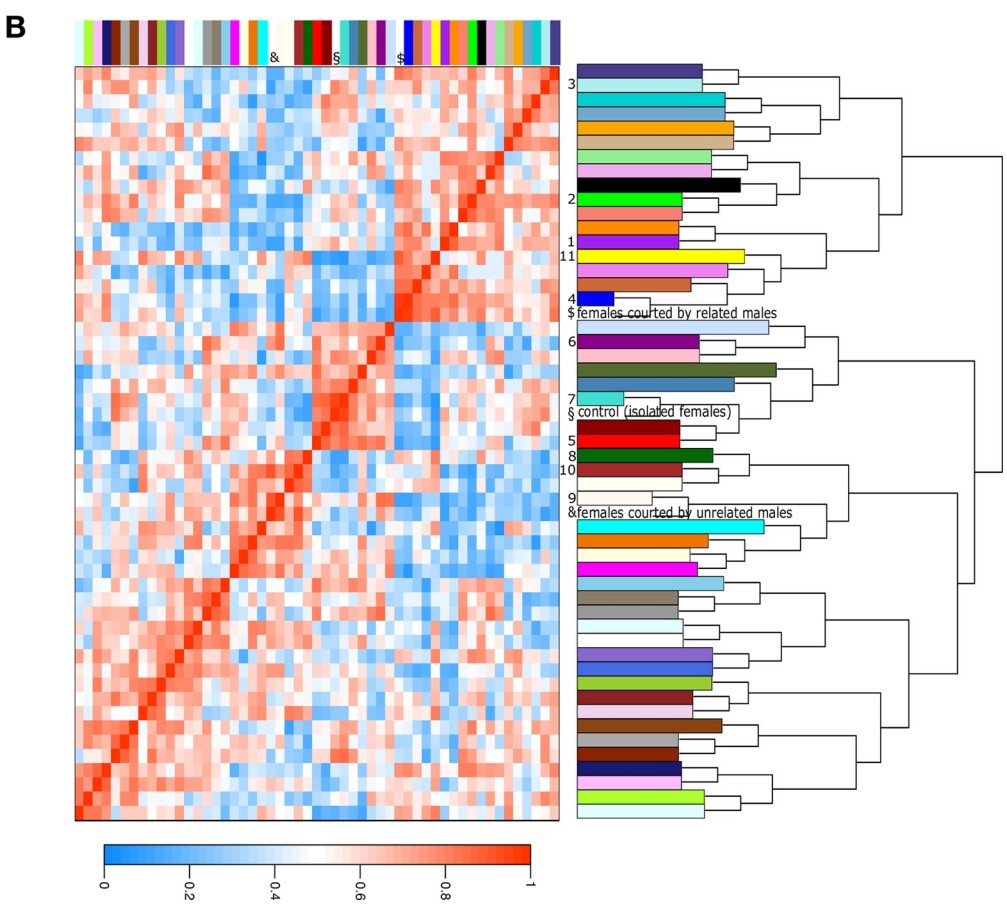

**Fig 4. Gene co-expression network of the full transcriptome (14,027 genes) in response to courtship and according to relatedness with the courting male.** (A) Clustering dendrogram of 14,027 genes (top side), with assignation to 50 modules of co-expressed genes represented by colors (bottom side). B) Summary plot of modules dendrogram and relationship with social environment experienced. The right panel shows the dendrogram of modules from gene network. The heatmap (left panel) illustrates pairwise correlation between modules of co-expressed genes and social environment: red denotes high positive correlation, white indicates no correlation, while blue indicates high negative correlation. The 11 modules significantly correlated with at least one of the social environment experienced by females were numbered from 1 to 11. Social environments were labelled with symbols: §; control (isolated females), $; females courted by related males, and &; females courted by unrelated males.

with a partner, regardless of the relatedness of the courting male. This suggests that transcriptomic shifts immediately arise following the male display, even before the eventual copulation. These transcriptomic changes can be triggered by the presence of a male, mate evaluation, or from a social encounter. With the current experimental protocol, it is not possible to distinguish the origin of these changes. The addition of two other controls such as females in the presence of i) an unrelated female or ii) a related female would allow further specification of the impact of social environment on neurogenomic responses. Overall, 7.1% of the total transcriptome was differentially expressed at most within ten minutes after the courtship started (1,001 DEGs). Such a neurogenomic response on the time scale of minutes following environmental change is mediated by immediate early genes [70]. For such genes, near-instantaneous transcription is allowed by the presence of RNA polymerase II, which stalls in the promoter regions of these genes [71].

A recent transcriptomic study conducted on female mate preference in the guppy (*Poecilia reticulata*) showed that the presence of a potential mate induced changes in the brain transcriptome after only 10 minutes of exposure [30]. In insects, few studies have attempted to identify transcriptional changes associated with courtship displays; to the best of our knowledge, all have focused on *Drosophila* [45, 46]. Immonen and Ritchie submitted *D. melanogaster* females to a courtship song diffused by a speaker for 15 minutes and found only 41 DEGs

**Table 1. Significant correlations between 11 modules and courtship experienced by females.**

| Module name | Module size | representative enriched GO terms | Controls (isolated females) | Females courted by related males | Females courted by unrelated males |
|---|---|---|---|---|---|
| 1 (purple) | 227 | molybdenum ion binding | NS | NS | -0.79 (0.02) |
| 2 (green) | 935 | nucleotide metabolic process | NS | NS | -0.73 (0.04) |
| 3 (pale turquoise) | 77 | mitochondrion, respiration | NS | NS | -0.8 (0.02) |
| 4 (blue) | 2108 | ribosome, sensory perception | NS | 0.94 (5e-04) | NS |
| 5 (red) | 643 | Methylation | NS | -0.76 (0.03) | NS |
| 6 (dark magenta) | 73 | GDP-mannose metabolic process | NS | -0.74 (0.04) | NS |
| 7 (turquoise) | 2675 | mating behaviour | 0.91 (0.002) | NS | NS |
| 8 (dark green) | 105 | cell surface receptor signaling pathway | -0.77 (0.03) | NS | NS |
| 9 (floral white) | 38 | oxidoreductase activity | NS | -0.8 (0.02) | 0.82 (0.01) |
| 10 (brown) | 1989 | response to stimulus, protein kinase activity | -0.73 (0.04) | NS | 0.81 (0.01) |
| 11 (yellow) | 1022 | carbohydrate catabolic process, metal ion binding | -0.8 (0.02) | 0.79 (0.02) | NS |

NS: not significant

Correlation coefficients and *P*-values in parentheses are indicated when significant (*P*-value <0.05). The results of enrichment analyses are indicated in the GO terms column (full lists in S4 Table).

(0.3% of the transcriptome) between courted and control females (in the presence of a male that was unable to perform a courtship display). Veltsos et al. (2017) submitted *D. pseudoobscura* females to male courtship and examined head transcriptomes immediately after mounting. They identified only 16 DEGs between courted and control females (virgin isolated females) (0.1% of the transcriptome). Together, these studies suggest that the female neurogenomic response to courtship only affects a very small set (>1%) of genes in Drosophila. Interestingly, in contrast with those studies, we found that the neurogenomic response to courtship was much greater in the wasp *V. canescens*. The differences may rely on the contrasted mating systems in these species. Despite the cost of reproduction, females mate multiple times in the majority of animal species in the wild including *D. melanogaster* [72] and *D. pseudoobscura* [73], most often with different males [74–76]. The benefits of polyandry for females include an adequate sperm supply [77], an increase in sperm competition [74] and a reduction in the cost of inbreeding [7]. In species with sl-CSD sex determination, mate choice is particularly determinant for female fitness, given the risk of genetic incompatibility. Furthermore, in monandrous species, such as *V. canescens* and a majority of parasitoid wasps (80%) [78], all progeny will have the same genitor. Thus, it is likely that selective pressures regarding mate choice should be stronger compared to those of polyandrous females. We suggest that the mating system might be an important determinant influencing the extent of neurogenomic response to courtship.

From a functional point of view, transcription factor activity and reproductive behaviour were some of the functions regulated following courtship. Indeed, we identified numerous transcription factors regulated following courtship, consistent with the large number of DEGs observed. In particular, we emphasized the transcription factors orthologous to *Thd1* and *muscleblind* and the kinase *Pink1*. All of these genes were also differentially expressed following courtship in *D. melanogaster* [45]. These candidates may have a conserved regulation pattern following courtship and might be associated with the response to courtship in insects. Our results also indicated the regulation of genes related to neurotransmitters, such as dopamine. The dopamine transporter *DAT* is overexpressed in courted females and the *ebony* gene involved in dopamine catabolism is downregulated in courted females, which is compatible with an increase in dopamine concentration following courtship. Dopamine is notably implicated in the control of motivation, movement and memory in the fruit fly *D. melanogaster* [79], and increased dopamine levels result in increased responsiveness to courtship cues [80]. Furthermore, we detected three neuropeptides downregulated in females courted by an unrelated male: *Capability*, *PDF* and *SIFamide*. These neuropeptides are involved in female sexual behaviour in the fruit fly, since reduction or absence of *SIFamide* makes females extremely receptive [81], while *PDF*-mutant females show an increased frequency of re-mating compared to wild-type females [82]. These three neuropeptides are candidates for involvement in the modulation of female receptivity. Together, these results demonstrate that the neurogenomic response to courtship in *V. canescens* involves neurotransmitters and neuropeptides. These genes are prime targets for further functional analyses.

Although courtship accounted for the majority of the detected DEGs, our experimental design nonetheless highlighted the major influence of relatedness between partners on female response to courtship. In total, 9.1% of the transcriptome was differentially expressed when comparing females courted by related males to those courted by unrelated males. In addition to the 2,780 courtship-responding genes that exhibited the same expression pattern whatever the relatedness of the courting male (modules 7 and 8), the network analysis highlighted 3 modules of sib-responsive genes (modules 4, 5 and 6; 2,824 genes) and 3 modules of genes regulated only in females courted by unrelated males (modules 1, 2 and 3; 1,239 genes) that could be related to female receptivity. Our results clearly showed that the relatedness of the courting male had a

major influence on the female response to courtship and that sib mating avoidance behaviour observed in this species is correlated to complex and massive changes in gene expression.

Concerning the functions associated with genes that vary according to the relatedness of the courting male, the GO terms Reproductive Behaviour and Male Mating Behaviour are notable. All genes underlying both terms are homologous to the *yellow—major royal jelly proteins* (*yellow-mrjp*). The *yellow* gene is unique to insects and some bacteria, while the *mrjp* gene family is restricted to Hymenopteran species and evolved from recent duplications of the *yellow* gene [83]. In the honey bee, the *mrjp* gene family is involved in both group social behaviour (royal jelly is constituted with 90% of MRJP proteins), and in individual sexual behaviour, with sex- and caste-specific gene expression patterns [81]. *Yellow-mrjp* functions in parasitic wasps are unknown, even though the largest expansion of the family described so far came from the Nasonia parasitic wasp genome, where genes are expressed broadly in different tissues and life stages [84]. In this study, we identified 6 members of the *yellow-mrjp* family that were DEGs following courtship, of which one presented differential expression according to the relatedness of the courting male. Further functional characterization will be required to test whether the *yellow-mrjp* gene family is involved in sib mating avoidance and female receptivity. Among the regulated genes, we also highlighted transcription factors such as *PAX6* that could drive the transcriptomic changes accompanying female mate choice. Very few studies have explored the molecular pathways underlying kin recognition. In the amphibian *Xenopus laevis*, tadpoles exhibited plasticity in social preferences according to exposure to kinship odourants [85]. Sustained kin odourants exposure during development drives changes in neurotransmitter expression from GABA to dopamine neurons, which are stimulated from an increase in the expression of the transcription factor *PAX6* and accompanied by a behavioural preference for kin odourants [85]. Here, we observed that the *V. canescens PAX6* orthologue is downregulated in females courted by brothers. Further studies to characterize *PAX6* function in *V. canescens*, particularly in the context of kin recognition, could test whether common molecular pathways could be elicited for kin recognition in distant taxa such as amphibians and insects.

We had formulated two non-mutually exclusive hypotheses. First, the perception of courtship was mediated by a change in gene expression, that would result in similar expression patterns in all females regardless of their relationship to the courting male. We identified such patterns for 2,780 genes (modules 7 and 8). Second, changes in female receptivity could result in changes in transcriptomic profiles. In this case, similar expression patterns would be expected for isolated females and females courted by their brothers. We have identified such expression patterns for 1,239 genes (modules 1, 2 and 3). Thus our results suggest that both courtship perception and changes in female receptivity induce a different neurogenomic response. In addition to sib mating avoidance, *V. canescens* females express kin recognition in the context of host choice, since females prefer to lay their eggs in a host parasitized by others than by a relative [48]. Neurogenomic analysis of responses to the presence of a relative in distinct ecological contexts would determine whether there are molecular bases associated with kin recognition.

## Supporting information

**S1 Fig. Exclusion of one biological replicate from the analysis due to outlier status.** Multivariate analysis based on the expression profiles of the 500 genes with the highest variance across all samples showed that one biological replicate corresponding to females courted by related males (2 empty circles representing 2 technical replicates) is far from the other points (full circles) A) in the plan defined by the two first axes of the principal component analysis and B) in the sample hierarchical clustering dendrogram. Significant statistical support of

outlier status is indicated by an approximately unbiased *p*-value with one million multiscale bootstrap replicates.
(TIF)

**S2 Fig. Boxplot showing expression patterns of the 6 members of the *yellow–mrjp* family were regulated following courting displays and/or according the relatedness between the female and the courting male.** Boxplot colours indicate biological condition: grey, isolated females; blue, females courted by unrelated males; orange, females courted by related males. The Y axes show the normalized counts after log transformation and VST normalization, and the boxplot whiskers show the range of reads between biological replicates. *, *P*-adj (FDR) < 0.01; NS, not significant.
(TIF)

**S3 Fig. Boxplots showing the expression patterns of the top 5 hub genes for each of the 11 modules varied according to experimental conditions.** The Y axes show the normalized counts after log transformation and VST normalization, and the boxplot whiskers show the range of reads between biological replicates.
(TIF)

**S1 Table. Summary statistics of the sequencing runs and transcriptome assembly.**
(XLSX)

**S2 Table.** Control females versus females courted by both related and unrelated males: A) list of 1,001 DEGs; B) 7 GO terms enriched in control females; C) 36 GO terms enriched in courted females.
(XLSX)

**S3 Table.** Females courted by related males versus females courted by unrelated males: A) list of 831 DEGs; B) 22 GO terms enriched in females courted by unrelated males; C) 66 GO terms enriched in females courted by related males.
(XLSX)

**S4 Table. Enriched GO terms associated with the 11 significant modules of co-expressed genes.**
(XLSX)

**S1 File.**
(CSV)

**S2 File.**
(CSV)

**S3 File.**
(TXT)

**S4 File.**
(CSV)

**S5 File.**
(R)

## Acknowledgments

The genome sequences of *Venturia canescens* were kindly provided by Dr. JM Drezen (IRBI, Univ. Tours, France). This work was performed using the computing facilities of the CC LBBE/PRABI.

## Author Contributions

**Conceptualization:** Aurore Gallot, Emmanuel Desouhant.

**Data curation:** Aurore Gallot.

**Formal analysis:** Aurore Gallot.

**Funding acquisition:** Aurore Gallot, Emmanuel Desouhant.

**Investigation:** Aurore Gallot, Sandrine Sauzet.

**Methodology:** Emmanuel Desouhant.

**Project administration:** Emmanuel Desouhant.

**Visualization:** Aurore Gallot.

**Writing – original draft:** Aurore Gallot.

**Writing – review & editing:** Aurore Gallot, Emmanuel Desouhant.

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
