## [Decision Letter · Decision Letter 0]

21 Feb 2020

PONE-D-20-01602

Kin recognition: neurogenomic response to mate choice and sib mating avoidance in a parasitic wasp

PLOS ONE

Dear Dr. Gallot,

Thank you for submitting your manuscript to PLOS ONE. After careful consideration, we feel that it has merit but does not fully meet PLOS ONE’s publication criteria as it currently stands. Therefore, we invite you to submit a revised version of the manuscript that addresses the points raised during the review process.

Two external referees and me have reviewed your manuscript and found some interesting results, but we agree that they paper needs major revisions and much clarification before possible publication. The reviewers have provided copious criticisms and comments. I will emphasize their major concerns of a rather small experimental design and sample sizes, and in some places very difficult to follow descriptions of the results. The main conclusions regarding gene modules and the actual transcriptional differences are not always well described and need to be further explained. One way to simplify the most important results would to be use a more conservative FDR P=0.01 instead of 0.05. Figure 4 is almost unreadable, and I agree the many of the figures need to be redrawn. The color schemes are not that helpful, and as one reviewer pointed out, are not likely to be viewer friendly for readers with color vision issues.

We would appreciate receiving your revised manuscript by Apr 06 2020 11:59PM. To enhance the reproducibility of your results, we recommend that if applicable you deposit your laboratory protocols in protocols.io, where a protocol can be assigned its own identifier (DOI) such that it can be cited independently in the future. For instructions see: http://journals.plos.org/plosone/s/submission-guidelines#loc-laboratory-protocols

We look forward to receiving your revised manuscript.

Kind regards,

William J. Etges

Academic Editor

PLOS ONE

Journal Requirements:

Reviewers' comments:

Reviewer's Responses to Questions

**Comments to the Author**

1. Is the manuscript technically sound, and do the data support the conclusions?

Reviewer #1: Partly

Reviewer #2: Partly

2. Has the statistical analysis been performed appropriately and rigorously? 

Reviewer #1: Yes

Reviewer #2: No

3. Have the authors made all data underlying the findings in their manuscript fully available?

Reviewer #1: No

Reviewer #2: Yes

4. Is the manuscript presented in an intelligible fashion and written in standard English?

Reviewer #1: Yes

Reviewer #2: Yes

5. Review Comments to the Author

**Reviewer #1:** The manuscript by Gallot and colleagues provides gene expression data from female heads that change in expression in response to courting and also that differ in their change in expression when courting is between related and unrelated parasitoid wasps. Females only mate once in this species, making the distinction between sib and non-sib very important, and the manuscript adds to phenotypic data showing females are able to distinguish kin and avoid mating with their brothers. Some good candidate genes whose expression changes accordingly are also reported.

One major issue with the paper is that the courtship assay is unclear. In particular, did all pairs produce courtship behaviour in each sample? It seems not from line 136, and line 103 mentions some females were not receptive. The experimental design thus seems to conflate receptivity with mating with sibs, since females have less receptivity to sibs. Was the proportion of successful courtship the same for sibs and non-sibs in each sample pool? Could the results be interpreted as a difference in harassment females experience between the treatments? A more detailed description of the assay is required in order to be able to comment on the discussion of the results.

In addition, the removal of one of the three biological replicates is problematic. The PCA analysis shows the samples have a more extreme profile towards the direction that their biological condition tends to. More discussion on the reasons the sample is considered unreliable is needed. Did less reads map to the reference? Was the RNA for this treatment collected at a different time of day than the other samples? More generally, the design of sample collection is unclear. Is it possible that circadian effects may have affected the design (for example if the courtship assays were performed at different times of the day or different days for the contrasted conditions?) Were the females from the same families in each pool, or is it possible that some conditions over-represent some families?

The GO enrichment tests were done against the full transcriptome (line 266). They should be done against the transcriptome with sufficient counts (i.e. only the 14,034 genes tested - 7 outliers that were excluded). This would be a more fair test.

I don't think it is a problem to avoid using a logFC threshold, but the authors should mention it explicitly in the text (line 240), and also provide some visualisation of how it would affect the results if one was employed. An easy way to do that is to provide a volcano plot (logCPM vs logFC) indicating the significant transcripts, so the reader can visualise the effect of any logFC threshold. The same figure would also be useful to illustrate the genes that are discussed in detail, such as the three neuropeptides in line 326.

Finally it would be good to provide the script and counts used for RNAseq analysis.

Overall I think this would be an interesting study to many readers. The main issue is to help interested readers compare the results to the rest of the literature. Both the provision of the script used for differential expression analysis, and a more detailed description of the conditions under which RNA was collected would help in comparing the results of this study to the literature.

Minor comments

Lines 71, 72, 76: sl-CSD -> sl-CSD locus (ie should indicate whether "sl-CSD" refers to the sex determination system or the genetic locus).

Line 79: please mention whether these experiments involved choice between two males, or not.

Please mention some statistics on the number of reads used in the RNAseq analysis, i.e. after filtering.

Line 321: should this number be 14,034 - 7?

Line 323: delete "levels", and clarify that 6.2% refers to tested gene set (I expected it to refer to the proportion 865/1,884)

Line 433: important -> significant

Line 436: I think the sentence "Thus ... state" is unnecessary as it repeats the message from previous sentences.

Line 447: delete "more precicely"

Line 451: within ten minutes -> "at most within 10 minutes". I think it is better to give a name to the condition captured, which is a better description of the biology e.g. "upon female acceptance of a mate" or "upon mounting by the male"

The red-green colour scheme in the figures is not colour-blind friendly.

I don't think panel A is needed for Fig 2 and Fig 3, the description in the text is sufficient. The legend is missing.

Fig 4: It is unclear what the "related control unrleated" text refers to in panel B. Also the text on the left part of panel B reads from top to bottom, while in other figures it reads from bottom to top. It would be best to rotate it by 180 degrees. More generally, please consider referring to the modules by a number rather than colours, which are too many to follow, especially for colour-blind readers.

This could be a major comment but I leave it to the authors to decide on whether it is worth following up. I think there is potential for an analysis of the interaction between genes that responded to courtship, and those that responded to courtship only with sibs or non-sibs.

For example one would make two virgin/courted contrasts: courted by sibs with virgins, and courted by non-sibs with virgins, and compare these results. There are many ways to illustrate this, for example

1. a scatterplot logFC virgin-sib courted vs logFC virgin -non sib courted or

2. 4-way venn: up courted with sib, up courted win non-sib, down courted with sib, down courted with non-sib.

This could indicate overall patterns of gene expression. For example are genes that are higher expressed when courted by sibs also expressed higher in courted by non-sibs, compared to virgins? Or do genes that are upregulated when mating with non-sibs resemble the expression pattern in virgins? This seems like the question in lines 102-104.

I appreciate Figure 4 is an alternative attempt to do this but if so it has not been discussed sufficiently to answer this question. One issue is that it is more difficult to appreciate the direction of change in gene expression in gene networks, but the information is there. At a minimum the authors could link the discussion of gene modules more directly with the original question in lines 102-104.

**Reviewer #2:** In general, this is an interesting study examining differential gene expression in females courted by related or unrelated males in a parasitoid wasp. This type of study is timely, and the authors do a good job presenting their research in the context of current work on both kin recognition and brain transcriptomics.

I am a little concerned about the small sample size (the authors have an N of 3 per treatment, plus technical replicates), and the authors apparent use of the technical replicates as independent samples. They need to be sure to pool the technical replicates for their statistical analyses, and to better articulate their actual sample size for their statistical analyses.

While their sample size is low, the results remain potentially interesting. I say potentially because I found the authors’ description of the different modules of genes associated with the different social conditions tested particularly difficult to read. The figures do not do a good job of illustrating what the clusters are, or how different gene expression is between the three treatments. The written description of these clusters was also difficult to read. This may be partially because the authors use the term “define” to describe these modules- but aren’t these modules of genes defined by the cluster analysis, not the authors themselves? Thus a better way to talk about these modules may be “X modules were associated with courtship….” That, coupled with a figure that better illustrates the different clusters, would make the results section seem less subjective. I suggest the authors revise their figures to show the different modules of genes that are expressed in these three different treatments. The authors should also describe which GO terms are associated with the different modules, not just state that there are GO terms associated with the modules.

A last major concern was that the authors were light on citations throughout their manuscript. I have mentioned a few of the many sentences missing citations in my minor comments below, but this is by no means exhaustive.

Minor/Specific points:

Line 37: Mates are also selected as a result of sensory bias (Fuller et al, 2005; Ryan & Cummings 2013). This hypothesis should be included in the introduction as well.

Line 59: missing a citation here.

Line 77: missing a citation here.

Line 79: missing a citation here.

Line 166: The number of biological replicates is quite low, particularly considering the number of treatments. That doesn’t mean the study isn’t useful, but does suggest a high likelihood of type II errors.

Line 204 and throughout: I suggest the authors clarify that they have 9 samples of 10 pooled heads each, plus 9 technical replicates. This is really a sample size of 9, with technical replicates, and should be analyzed accordingly. The technical replicates are not independent, thus the authors do not have a sample size of 18. Since these are technical replicates instead of independent individuals, they may skew the results, as they will amplify the differences between the three individuals in each treatment, and give an illusion of low within-treatment variance.

Lines 220-221: It is particularly disconcerting that the authors excluded one of their three samples from the sib-courting treatment. This brings the sample size for this treatment down to two, which is really too small for statistical analyses. Given that each of these samples contains the RNA from 10 heads, this sample dissimilarity suggests to me there was an error in the RNA extraction. However, an N of 2 is really too small for statistical analyses.

Line 283: Only 37.5% of the genes were most similar to Hymenopteran proteins? What were the rest of the genes most similar to? Drosophila or other insect proteins? A more detailed description of the gene set would be appreciated here.

Line 284: Only 33.4% of the genes were associated with at least one GO term? Is that because most of the genes were uncharacterized? Please clarify.

Line 288: What happens when the authors run these analyses with those outliers? Do most of the outlier reads map back to Hymenoptera? Or are they something else?

Line 291: Please define the groups by treatment and not color in the text (color should be in the figure legend, and remember, some readers will print this paper in black and white).

Line 384-388: The explanation of the visualization of these 50 different modules should be in the figure legend, not the main text.

Line 392: Naming the modules by colors is not particularly descriptive. Are there more biologically relevant names the authors could use for these modules?

Line 428: citation needed here.

Line 522: The genes constitute 90% of the royal jelly? Or are expression patterns of these genes responsible for 90% of the production and content of the royal jelly?

Line 538: Citation needed here.

Figure 4: The trees in panel A for figure 4 are so dense they are not readable. I suggest either removing them or changing the thickness of the lines.

6. PLOS authors have the option to publish the peer review history of their article (what does this mean?). If published, this will include your full peer review and any attached files.

Reviewer #1: No

Reviewer #2: No

---

## [Author Response · Author response to Decision Letter 0]

7 Aug 2020

5. Review Comments to the Author

Reviewer #1: The manuscript by Gallot and colleagues provides gene expression data from female heads that change in expression in response to courting and also that differ in their change in expression when courting is between related and unrelated parasitoid wasps. Females only mate once in this species, making the distinction between sib and non-sib very important, and the manuscript adds to phenotypic data showing females are able to distinguish kin and avoid mating with their brothers. Some good candidate genes whose expression changes accordingly are also reported.

One major issue with the paper is that the courtship assay is unclear. In particular, did all pairs produce courtship behaviour in each sample? It seems not from line 136, and line 103 mentions some females were not receptive. The experimental design thus seems to conflate receptivity with mating with sibs, since females have less receptivity to sibs. Was the proportion of successful courtship the same for sibs and non-sibs in each sample pool? Could the results be interpreted as a difference in harassment females experience between the treatments? A more detailed description of the assay is required in order to be able to comment on the discussion of the results.

-The courtship behavior, i.e. a courtship parade produced by the male nearby the female, has been observed in all the pairs included in the study. Females did not experienced harassment by males, since males produce only a maximum of one courtship parade sequence. Whenever this behaviour sequence was not observed, the couple was eliminated from the study. 

This information has been added to the manuscript (l. 147) “The 10 minutes period coincides with an active male courtship behaviour(42). For each female under conditions 1 and 2, one sequence of active courtship parade was observed within the 10 minutes following the introduction of the male to the box for all pairs. Whenever this behaviour sequence was not observed, the pair was eliminated from the study.” 

Female receptivity has not been assessed the in this study. Therefore, the results on female receptivity (i.e. the increase in female receptivity in presence of unrelated males compared to in the presence of a related males) came from a study previously published by Metzger et al. (2010). 

In addition, the removal of one of the three biological replicates is problematic. The PCA analysis shows the samples have a more extreme profile towards the direction that their biological condition tends to. More discussion on the reasons the sample is considered unreliable is needed. Did less reads map to the reference? Was the RNA for this treatment collected at a different time of day than the other samples? More generally, the design of sample collection is unclear. Is it possible that circadian effects may have affected the design (for example if the courtship assays were performed at different times of the day or different days for the contrasted conditions?) Were the females from the same families in each pool, or is it possible that some conditions over-represent some families?

-Concerning the removal of one sample (brother, replicate 1), we were unable to establish any difference explaining the extreme profile of this peculiar sample neither during sample collection neither in the data mapping process. The hypothesis that seems most plausible to us is that a problem occurred during the extraction of RNA for this peculiar sample. However, we were not able to identify differences in the reads mapped to the reference (one-way anova, p=0.895). 

This sample, just as all the other is constituted of a pool of individual heads, and every individual in every samples have been randomized in the same way: “The experimental design has been conceived in order to minimize the influence of circadian rhythm and genetic background on the results. Newly emerged females were isolated every morning and numbered. A random draw was then made to establish the order of passage of the females and the treatment assigned. Behavioural experiments took place in the afternoon between 1:00 and 4:00 p.m. for all females that were captured in the morning in the order established by the random draw. A maximum of 2 females from each family have been kept daily, and randomized in one condition. For each given condition, the ten females belonged to different families to avoid an effect of genetic homogeneity on transcriptomic results. New families have been produced for every biological replicate.” We have now added these precisions in this second version of the manuscript (l. 138). 

The GO enrichment tests were done against the full transcriptome (line 266). They should be done against the transcriptome with sufficient counts (i.e. only the 14,034 genes tested - 7 outliers that were excluded). This would be a more fair test.

-Thanks for this relevant suggestion. We have performed a new set of GO enrichment tests using the 14,027 transcripts (i.e. 14,034 minus 7 outliers) as a reference. All the results are included in supplementary tables, and the manuscript has been modified accordingly to these new results. These changes did not result in any major qualitative changes in our key results, which gives us confidence in the robustness of these results.

I don't think it is a problem to avoid using a logFC threshold, but the authors should mention it explicitly in the text (line 240), and also provide some visualisation of how it would affect the results if one was employed. An easy way to do that is to provide a volcano plot (logCPM vs logFC) indicating the significant transcripts, so the reader can visualise the effect of any logFC threshold. The same figure would also be useful to illustrate the genes that are discussed in detail, such as the three neuropeptides in line 326.

-We have added an explicit mention that we didn’t used a FC threshold (l. 259). We have also provided volcano plots for the 2 contrasts (figures 2A and 3A) where we have highlighted the genes discussed in detail.

Finally it would be good to provide the script and counts used for RNAseq analysis.

Overall I think this would be an interesting study to many readers. The main issue is to help interested readers compare the results to the rest of the literature. Both the provision of the script used for differential expression analysis, and a more detailed description of the conditions under which RNA was collected would help in comparing the results of this study to the literature.

-Raw data count table and R script have been added in supplementary material.

Minor comments

Lines 71, 72, 76: sl-CSD -> sl-CSD locus (ie should indicate whether "sl-CSD" refers to the sex determination system or the genetic locus).

-In the paragraph from l. 66 to l. 79 we have rephrased and indicated “sl-CSD” when it refers to the sex determination system and “sl-CSD locus” when it refers to the genetic locus.

Line 79: please mention whether these experiments involved choice between two males, or not.

-These experiments included the proportion of successful mate when a single female is in presence of sibs (i.e. presence of 2 brothers), or in presence of non sibs (i.e. presence of 2 unrelated males). Experiments involving choice between 2 males, one brother and one unrelated present in a same area have shown that female mates indifferently with the 2 males (Metzger et al., 2010). These points has now been explained lines 80-86.

Please mention some statistics on the number of reads used in the RNAseq analysis, i.e. after filtering.

-An overview of the RNAseq analysis statistics are summarized in the table S1. We have added some of those statistics in the results (l. 300): “After quality filtering a mean of 98.4% of paired-reads were kept, on which 70.8% were successfully mapped to the genome (representing a mean of 20 millions per sample, for a total of 363 million paired-reads, S1 table). Such values corresponded to the high-quality standards observed in other Hymenopteran species with an annotated genome(62).”

Line 321: should this number be 14,034 - 7?

-Indeed, we have corrected with the value 14,027 (l. 343)

Line 323: delete "levels", and clarify that 6.2% refers to tested gene set (I expected it to refer to the proportion 865/1,884)

-We have done the following modification l. 344-349 : “Among the 1,001 DEGs, 463 had higher expression in isolated females (3.3% of total transcriptome), gene ontology enrichment analysis reveals that this set of gene was enriched in DNA-binding Transcription Factor Activity (full list in S2 table). In contrast, 538 DEGs were overexpressed in courted females (3.8% of total transcriptome), gene ontology enrichment analysis reveals that this set of gene was enriched in Reproductive Behaviour (full list in S2 table).”

Line 433: important -> significant

-We have done the modification (l. 451)

Line 436: I think the sentence "Thus ... state" is unnecessary as it repeats the message from previous sentences.

-The sentence has been deleted (l. 454)

Line 447: delete "more precicely"

-We have done the modification (l. 467)

Line 451: within ten minutes -> "at most within 10 minutes". I think it is better to give a name to the condition captured, which is a better description of the biology e.g. "upon female acceptance of a mate" or "upon mounting by the male"

-The correction has been done (l. 470)

The red-green colour scheme in the figures is not colour-blind friendly.

-We have modified the color palette of all figures, using a colour-blind friendly palette (with blue, orange and grey colors).

I don't think panel A is needed for Fig 2 and Fig 3, the description in the text is sufficient. The legend is missing.

-Figures 2 and 3 have been modified by removing the A panels, that have been replaced by the volcano plots as suggested above. The legend is now included within the results section, in accordance with Plos one requirements.

Fig 4: It is unclear what the "related control unrleated" text refers to in panel B. Also the text on the left part of panel B reads from top to bottom, while in other figures it reads from bottom to top. It would be best to rotate it by 180 degrees. More generally, please consider referring to the modules by a number rather than colours, which are too many to follow, especially for colour-blind readers.

-Figure 4 has been modified “related” was replaced by “females courted by related males”; “unrelated” by “females courted by unrelated males”; and “control” by “control (isolated females)”. We also replaced this text previously located on the top of the heatmap, by the symbols “&”, “§” and “$” to increase clarity. Colours associated to modules have been conserved since the network illustration is based on colours attribution, but we considered the use of number rather than colour name, by numbering the all significant modules from 1 to 11. Then we refer to the gene module all along the manuscript only by using the numbered nomenclature. Thus, the manuscript as well as the figure 4 are now easier to follow, including for colour-blind readers. 

This could be a major comment but I leave it to the authors to decide on whether it is worth following up. I think there is potential for an analysis of the interaction between genes that responded to courtship, and those that responded to courtship only with sibs or non-sibs.

For example one would make two virgin/courted contrasts: courted by sibs with virgins, and courted by non-sibs with virgins, and compare these results. There are many ways to illustrate this, for example

1. a scatterplot logFC virgin-sib courted vs logFC virgin -non sib courted or

2. 4-way venn: up courted with sib, up courted win non-sib, down courted with sib, down courted with non-sib.

This could indicate overall patterns of gene expression. For example are genes that are higher expressed when courted by sibs also expressed higher in courted by non-sibs, compared to virgins? Or do genes that are upregulated when mating with non-sibs resemble the expression pattern in virgins? This seems like the question in lines 102-104.

I appreciate Figure 4 is an alternative attempt to do this but if so it has not been discussed sufficiently to answer this question. One issue is that it is more difficult to appreciate the direction of change in gene expression in gene networks, but the information is there. At a minimum the authors could link the discussion of gene modules more directly with the original question in lines 102-104.

-Indeed, the purpose of the network analysis and the figure 4 (illustrating this analysis) is precisely to bring insight to this point. We are convinced that in the context of this study, such network approach is much more powerful than a set of bilateral comparisons. We considered this comment, together with the ones from the other reviewer and editor, and we understand that the results of this analysis need to be clarified in this manuscript. Thus we have proposed in this revised version of the manuscript a range of modifications accordingly:

o a new version of the figure 4 to illustrate the network analysis. 

o a new description of the result paragraph

o modifications of the discussion directly oriented to the original question in the introduction (l. 102-104)

Concerning the last point, a part of the discussion has been added (l. 563-571):“We had formulated two non-mutually exclusive hypotheses. First, the perception of courtship was mediated by a change in gene expression, that would result in similar expression patterns in all females regardless of their relationship to the courting male. We identified such patterns for 2,780 genes (modules 7 and 8). Second, changes in female receptivity could result in changes in transcriptomic profiles. In this case, similar expression patterns would be expected for isolated females and females courted by their brothers. We have identified such expression patterns for 1,239 genes (modules 1, 2 and 3). Thus our results suggest that both courtship perception and changes in female receptivity induce a different neurogenomic response.”

Reviewer #2: In general, this is an interesting study examining differential gene expression in females courted by related or unrelated males in a parasitoid wasp. This type of study is timely, and the authors do a good job presenting their research in the context of current work on both kin recognition and brain transcriptomics.

I am a little concerned about the small sample size (the authors have an N of 3 per treatment, plus technical replicates), and the authors apparent use of the technical replicates as independent samples. They need to be sure to pool the technical replicates for their statistical analyses, and to better articulate their actual sample size for their statistical analyses.

-It seems we were not clear enough in the first version, since we did not used the technical replicates as independent samples. The technical replicates have been used independently only for the PCA and hierarchical clustering of libraries (figure 1). Then, each pair of technical replicates has been merged before proceeding to differential expression analysis and network analysis, as recommended. We have now added precisions on sample size in the material and methods (l. 217-222 and l. 241-244).

While their sample size is low, the results remain potentially interesting. I say potentially because I found the authors’ description of the different modules of genes associated with the different social conditions tested particularly difficult to read. The figures do not do a good job of illustrating what the clusters are, or how different gene expression is between the three treatments. The written description of these clusters was also difficult to read. This may be partially because the authors use the term “define” to describe these modules- but aren’t these modules of genes defined by the cluster analysis, not the authors themselves? Thus a better way to talk about these modules may be “X modules were associated with courtship….” That, coupled with a figure that better illustrates the different clusters, would make the results section seem less subjective. I suggest the authors revise their figures to show the different modules of genes that are expressed in these three different treatments. The authors should also describe which GO terms are associated with the different modules, not just state that there are GO terms associated with the modules.

-We understand that the results of this analysis need to be clarified in the manuscript.

o The description of the different modules of genes associated with the different social conditions has been modified following the reviewer suggestions (l. 400-419).

o The figure 4 illustrating the gene network analysis has been modified: i) the gene tree and gene network in the panel A have been changed in order to improve the clarity of how the clusters have been defined; ii) in the panel B significant modules have been numbered from 1 to 11; iii) the name of the social conditions have been modified.

o Some enriched GO terms associated with the significant modules have been highlighted in the table 1 within the manuscript, but exhaustive lists are still contained in supplementary table 4.

A last major concern was that the authors were light on citations throughout their manuscript. I have mentioned a few of the many sentences missing citations in my minor comments below, but this is by no means exhaustive.

-The revised version of the manuscript has been enriched with numerous citations throughout the manuscript.

Minor/Specific points:

Line 37: Mates are also selected as a result of sensory bias (Fuller et al, 2005; Ryan & Cummings 2013). This hypothesis should be included in the introduction as well.

-We have added this hypothesis in the introduction (l. 37): “The ‘good genes’ hypothesis predicts that females favour reproduction with males carrying traits that are honest indicators of good genes or as a result of sensory bias(3,4), hence obtaining genetic benefits for their offspring(5).”

Line 59: missing a citation here.

-A citation has been added here (l. 59-63) “Transcriptomic studies provided recent insight into female mating decisions(27). Coordinated changes in the expression of many genes in female brains, i.e., a neurogenomic response, have been identified following courtship in Poeciliidae fishes(28–30). This response depends on male attractiveness and is in accordance with female preferences.”

Line 77: missing a citation here.

-A citation has been added here (l. 77-79): “In the parasitoid wasp Venturia canescens, which has sl-CSD(38), females only mate once(39), making mate choice particularly decisive.”

Line 79: missing a citation here.

-A citation has been added here (l. 79-80): “Indeed, in this species, females are able to discriminate kin and non-kin during male courtship based on olfactory-mediated cues(40).”

Line 166: The number of biological replicates is quite low, particularly considering the number of treatments. That doesn’t mean the study isn’t useful, but does suggest a high likelihood of type II errors.

Line 204 and throughout: I suggest the authors clarify that they have 9 samples of 10 pooled heads each, plus 9 technical replicates. This is really a sample size of 9, with technical replicates, and should be analyzed accordingly. The technical replicates are not independent, thus the authors do not have a sample size of 18. Since these are technical replicates instead of independent individuals, they may skew the results, as they will amplify the differences between the three individuals in each treatment, and give an illusion of low within-treatment variance.

-We have clarified the number of samples (9 biological samples), each represented by 2 technical replicates (18 transcriptomic libraries) in material and methods. The 18 libraries were used only for PCA and hierarchical clustering. Then technical replicates have been merged for differential expression test and network analysis. We have now precisions on these points in the material and methods (l. 217-222 and l. 241-244).

Lines 220-221: It is particularly disconcerting that the authors excluded one of their three samples from the sib-courting treatment. This brings the sample size for this treatment down to two, which is really too small for statistical analyses. Given that each of these samples contains the RNA from 10 heads, this sample dissimilarity suggests to me there was an error in the RNA extraction. However, an N of 2 is really too small for statistical analyses.

-Concerning the removal of one sample (brother, replicate 1), we also speculate that the more probable explanation is that a problem occurred during RNA extraction. However, we were unable to establish any difference explaining the extreme profile of this peculiar sample neither during sample collection neither in the data mapping process (see our reply to a comment of referee 1). There were no differences in the proportion of reads mapped to the reference (one-way anova, p=0.895). This sample, just as all the other is constituted of a pool of 10 individual heads, and every individual in every samples have been randomized in the same way.

We are aware of the limitations related to the small sample size. We now clearly mentioned this limit in the discussion (l. 454-460) : “Despite the quite low number of biological replicates, the highly contrasted transcriptomes observed in the different social contexts suggest that sib mating avoidance behaviour could be considered a neurogenomic state. This research paves the way for further study on neurogenomic effects of sib mating avoidance in many species where such behaviours have been described and, thus, may contribute to the understanding of the molecular mechanisms underlying the evolution of avoiding consanguinity.”

Line 283: Only 37.5% of the genes were most similar to Hymenopteran proteins? What were the rest of the genes most similar to? Drosophila or other insect proteins? A more detailed description of the gene set would be appreciated here.

-The reviewer pointed a mistake in the data that has been now corrected in this version of the manuscript (l.337): “Overall, 76% of predicted genes get a blast hit (12,740) while 4,012 sequences get no hit. Among genes matching with blast, 89.4% presented their best hit with an insect sequence, of which 84.4% match more specifically to a hymenopteran insect sequences.”

Line 284: Only 33.4% of the genes were associated with at least one GO term? Is that because most of the genes were uncharacterized? Please clarify.

-Indeed only 33.4% of the predicted genes in V. canescens genome get at least one GO annotation. This means that while a majority (76%) of predicted genes shown homology with protein sequences that have been previously deposited in databases, i.e., get a blast hit; only a low proportion of these sequences are associated to functional annotation (molecular function, cellular component, biological process). Such gene biological identity are scarce and essentially come from experiments performed on model organisms. Most of the genes were uncharacterized, notably in non-model metazoan species. Nonetheless, this proportion is rapidly increasing. In comparison, in human, only 32% of genes got at least one GO annotation in 2004; but in 2015, 65% of genes got at least one GO annotation (Tomcsak et al., 2018).

We have added precision in the manuscript (l. 308): “Finally, most of the genes were uncharacterized, since only 33.4% of the predicted genes (5,589) were associated with at least one Gene Ontology (GO) functional annotation.”

Line 288: What happens when the authors run these analyses with those outliers? Do most of the outlier reads map back to Hymenoptera? Or are they something else?

-We run again these analyses without those 7 outliers, since the 7 corresponding sequences do not match with any known sequences. All the results relative to differential expression and network analysis in this new version of the manuscript have been done on the dataset containing 14,027 genes.

Line 291: Please define the groups by treatment and not color in the text (color should be in the figure legend, and remember, some readers will print this paper in black and white).

-The correction has been done (l. 315)

Line 384-388: The explanation of the visualization of these 50 different modules should be in the figure legend, not the main text.

-The correction has been done (l. 408)

Line 392: Naming the modules by colors is not particularly descriptive. Are there more biologically relevant names the authors could use for these modules?

-We have now proposed to use number to characterize modules rather than color. Particularly, we have numbered from 1 to 11 the modules significantly correlated with at least one experimental condition, while other modules which are not mentioned in the manuscript have not been numbered. However, color names have been conserved in parenthesis, because the colors are used in the figure 4 to illustrate the gene network.

Line 428: citation needed here.

-A citation has been added here (l. 447) “I In V. canescens females, mate relatedness influences female sexual receptivity and is estimated during male courtship displays through chemical cues(40).”

Line 522: The genes constitute 90% of the royal jelly? Or are expression patterns of these genes responsible for 90% of the production and content of the royal jelly?

-This sentence was incorrect and has been rectified (l. 544): “royal jelly is constituted with 90% of MRJP proteins”.

Line 538: Citation needed here.

-A citation has been added here (l. 557): “Sustained kin odourants exposure during development drives changes in neurotransmitter expression from GABA to dopamine neurons, which are stimulated from an increase in the expression of the transcription factor PAX6 and accompanied by a behavioural preference for kin odourants(83).”

Figure 4: The trees in panel A for figure 4 are so dense they are not readable. I suggest either removing them or changing the thickness of the lines.

-We proposed a new illustration of the gene network in the figure 4. The tree panel resolution has been increased.

---

## [Decision Letter · Decision Letter 1]

7 Sep 2020

PONE-D-20-01602R1

Kin recognition: neurogenomic response to mate choice and sib mating avoidance in a parasitic wasp

PLOS ONE

Dear Dr. gallot,

Thank you for submitting your manuscript to PLOS ONE.  We invite you to submit a revised version of the manuscript that addresses the points raised during the review process.

I agree with both external referees that the authors have done a solid job of responding to the reviewers' comments and criticisms. Both reviewers have suggested a few more issues that need addressing before a final decision can be made.  I think the MS is much clearer and the results are better described for your readers.

We look forward to receiving your revised manuscript.

Kind regards,

William J. Etges

Academic Editor

PLOS ONE

Reviewers' comments:

Reviewer's Responses to Questions

**Comments to the Author**

1. If the authors have adequately addressed your comments raised in a previous round of review and you feel that this manuscript is now acceptable for publication, you may indicate that here to bypass the “Comments to the Author” section, enter your conflict of interest statement in the “Confidential to Editor” section, and submit your "Accept" recommendation.

Reviewer #1: All comments have been addressed

Reviewer #2: (No Response)

2. Is the manuscript technically sound, and do the data support the conclusions?

Reviewer #1: Yes

Reviewer #2: Yes

3. Has the statistical analysis been performed appropriately and rigorously? 

Reviewer #1: Yes

Reviewer #2: Yes

4. Have the authors made all data underlying the findings in their manuscript fully available?

Reviewer #1: Yes

Reviewer #2: Yes

5. Is the manuscript presented in an intelligible fashion and written in standard English?

Reviewer #1: Yes

Reviewer #2: Yes

6. Review Comments to the Author

Reviewer #1: The authors have responded to all my comments in a satisfactory manner.

There is one exception with regards to the sample that was removed. The text should mention why the sample represents a problem during preparation, and not due to biological variation, given that both technical replicates are very similar, and both are outliers.

Minor comments

Line 259:did not applied -> did not apply

Line 289: this should not be the full transcriptome, but the part of the transcriptome that was included in the analysis. The authors mention that they corrected for this, so I suspect this is text that has not been updated.

Lines 356-358: Please mention in the text the direction of high expression for the genes that are discussed.

Line 425: Showed -> shows, illustrated -> illustrates, denoted -> denotes, indicated -> indicates

Line 480: Veltos -> Veltsos

Fig 3A: Please check the top left outlier gene. Is it reliable? If not consider removing it to shorten the y axis.

Fig 4: the symbols and numbers overap, please use smaller text.

Reviewer #2: This is a much improved version of a previously reviewed manuscript: “Kin recognition: neurogenomic response to mate choice and sib mating avoidance in a parasitic wasp”. I have a few remaining minor comments that the authors should address.

Minor points:

Line 241: “All pairs of technical replicates were merged before….” Not “has been merged”.

Line 249: missing the words “of all” between expression and transcripts. It should read: “tested for differential expression of all transcripts with an average level of expression above 10 reads per library”

Figure 1 legend: Please state which color represents which treatment in the figure legend instead of stating that treatments are represented by different colors.

Line 381: This section is a little unclear. Are all 22 of these GO terms only associated with 2 genes? Or are they associated with some of the other 479 differentially expressed genes as well?

Lines 549-550: The authors need to mention the decades of research on the molecular pathways involved in kin-recognition in Hydractinia symbiolongicarpus and Botryllus schosseri. This topic has been well studied in colonial marine invertebrates since at least 1994 (Mokady & Buss, 1994; Fagan & Weissman, 1997; Cadavid et al., 2004; Litman, 2006; Nicotra et al., 2009; Nydam et al., 2017, to name a few of the many papers on this topic).

It looks like the authors included both the original 4 figures and the revised figures, but failed to mention which figures were the old figures and which figures were the new figures. It looks like the second set of figures are the new figures. Assuming that is true, the color scheme is definitely more color-blind friendly, though Figure 4b is still a little difficult to read. I suggest stretching the y-axis so the cluster numbers are easier to match to bars.

The new S1 figure is useful, as is S3. I appreciate their addition.

7. PLOS authors have the option to publish the peer review history of their article (what does this mean?). If published, this will include your full peer review and any attached files.

Reviewer #1: No

Reviewer #2: No

---

## [Author Response · Author response to Decision Letter 1]

5 Oct 2020

Dear William J. Etges,

We want to thank the reviewers as well as the editor for their corrections. We hope that this revised version includes all the changes requested.

A comprehensive response to the specific items provided by the 2 reviewers has been addressed. We have responded to each item in details in the following section (part 6).

Aurore Gallot, PhD.

PONE-D-20-01602R1

Kin recognition: neurogenomic response to mate choice and sib mating avoidance in a parasitic wasp

PLOS ONE

Dear Dr. gallot,

Thank you for submitting your manuscript to PLOS ONE. We invite you to submit a revised version of the manuscript that addresses the points raised during the review process.

I agree with both external referees that the authors have done a solid job of responding to the reviewers' comments and criticisms. Both reviewers have suggested a few more issues that need addressing before a final decision can be made. I think the MS is much clearer and the results are better described for your readers.

If applicable, we recommend that you deposit your laboratory protocols in protocols.io<http://protocols.io/> to enhance the reproducibility of your results. Protocols.io<http://protocols.io/> assigns your protocol its own identifier (DOI) so that it can be cited independently in the future. For instructions see: http://journals.plos.org/plosone/s/submission-guidelines#loc-laboratory-protocols

We look forward to receiving your revised manuscript.

Kind regards,

William J. Etges

Academic Editor

PLOS ONE

Reviewers' comments:

Reviewer's Responses to Questions

Comments to the Author

1. If the authors have adequately addressed your comments raised in a previous round of review and you feel that this manuscript is now acceptable for publication, you may indicate that here to bypass the “Comments to the Author” section, enter your conflict of interest statement in the “Confidential to Editor” section, and submit your "Accept" recommendation.

Reviewer #1: All comments have been addressed

Reviewer #2: (No Response)

2. Is the manuscript technically sound, and do the data support the conclusions?

Reviewer #1: Yes

Reviewer #2: Yes

3. Has the statistical analysis been performed appropriately and rigorously?

Reviewer #1: Yes

Reviewer #2: Yes

4. Have the authors made all data underlying the findings in their manuscript fully available?

The PLOS Data policy<http://www.plosone.org/static/policies.action#sharing> requires authors to make all data underlying the findings described in their manuscript fully available without restriction, with rare exception (please refer to the Data Availability Statement in the manuscript PDF file). The data should be provided as part of the manuscript or its supporting information, or deposited to a public repository. For example, in addition to summary statistics, the data points behind means, medians and variance measures should be available. If there are restrictions on publicly sharing data—e.g. participant privacy or use of data from a third party—those must be specified.

Reviewer #1: Yes

Reviewer #2: Yes

5. Is the manuscript presented in an intelligible fashion and written in standard English?

Reviewer #1: Yes

Reviewer #2: Yes

6. Review Comments to the Author

Reviewer #1: The authors have responded to all my comments in a satisfactory manner.

There is one exception with regards to the sample that was removed. The text should mention why the sample represents a problem during preparation, and not due to biological variation, given that both technical replicates are very similar, and both are outliers.

We explicitly added in the manuscript why the outlier sample is problematic and has been removed (l. 248-249).

Minor comments

All the minor comments has been taken in account in the new version of the manuscript

Line 259:did not applied -> did not apply

We have corrected (l. 271).

Line 289: this should not be the full transcriptome, but the part of the transcriptome that was included in the analysis. The authors mention that they corrected for this, so I suspect this is text that has not been updated.

We have corrected (l. 302).

Lines 356-358: Please mention in the text the direction of high expression for the genes that are discussed.

The direction of high expression is now described for all the discussed genes. We added this information for the genes ebony, DAT, 5-HT1A and serotonin receptor 1A (l. 372-375).

Line 425: Showed -> shows, illustrated -> illustrates, denoted -> denotes, indicated -> indicates

We have corrected (l. 444-447).

Line 480: Veltos -> Veltsos

We have corrected (l. 505).

Fig 3A: Please check the top left outlier gene. Is it reliable? If not consider removing it to shorten the y axis.

We controlled the outlier gene but we do not considered that it is aberrant given that the corresponding sequence encode and open reading frame that perfectly match to the mitochondrial gene CO1. Considering that the gene annotation is reliable, we did not remove the outlier point, and did not rescale the figure either.

Fig 4: the symbols and numbers overap, please use smaller text.

We stretched the y axis and used a smaller text in the figure 4b.

Reviewer #2: This is a much improved version of a previously reviewed manuscript: “Kin recognition: neurogenomic response to mate choice and sib mating avoidance in a parasitic wasp”. I have a few remaining minor comments that the authors should address.

Minor points:

Line 241: “All pairs of technical replicates were merged before….” Not “has been merged”.

We have corrected (l. 252).

Line 249: missing the words “of all” between expression and transcripts. It should read: “tested for differential expression of all transcripts with an average level of expression above 10 reads per library”

We have corrected (l. 261).

Figure 1 legend: Please state which color represents which treatment in the figure legend instead of stating that treatments are represented by different colors.

We added which color represent which treatment in the figure legend (l. 350-351). 

Line 381: This section is a little unclear. Are all 22 of these GO terms only associated with 2 genes? Or are they associated with some of the other 479 differentially expressed genes as well?

Only the 2 GO ‘Reproductive Behaviour’ and ‘Male Mating Behaviour’ are associated with 2 genes. We clarified this section (l. 401). 

Lines 549-550: The authors need to mention the decades of research on the molecular pathways involved in kin-recognition in Hydractinia symbiolongicarpus and Botryllus schosseri. This topic has been well studied in colonial marine invertebrates since at least 1994 (Mokady & Buss, 1994; Fagan & Weissman, 1997; Cadavid et al., 2004; Litman, 2006; Nicotra et al., 2009; Nydam et al., 2017, to name a few of the many papers on this topic).

We mentioned research on kin recognition in colonial invertebrates in the introduction (l. 47-51).

It looks like the authors included both the original 4 figures and the revised figures, but failed to mention which figures were the old figures and which figures were the new figures. It looks like the second set of figures are the new figures. Assuming that is true, the color scheme is definitely more color-blind friendly, though Figure 4b is still a little difficult to read. I suggest stretching the y-axis so the cluster numbers are easier to match to bars.

We stretched the y axis and used a smaller text in the figure 4b.

The new S1 figure is useful, as is S3. I appreciate their addition.

---

## [Editor Report · Decision Letter 2]

9 Oct 2020

Kin recognition: neurogenomic response to mate choice and sib mating avoidance in a parasitic wasp

PONE-D-20-01602R2

Dear Dr. gallot,

We’re pleased to inform you that your manuscript has been judged scientifically suitable for publication and will be formally accepted for publication once it meets all outstanding technical requirements.

Kind regards,

William J. Etges

Academic Editor

PLOS ONE
---

## [Editor Report · Acceptance letter]

16 Oct 2020

PONE-D-20-01602R2 

Kin recognition: neurogenomic response to mate choice and sib mating avoidance in a parasitic wasp 

Dear Dr. gallot:

I'm pleased to inform you that your manuscript has been deemed suitable for publication in PLOS ONE. Congratulations! Your manuscript is now with our production department. 

Kind regards, 

on behalf of

Dr. William J. Etges 

Academic Editor

PLOS ONE